



# Liquid–liquid phase separation in organic particles consisting of α-pinene and β-caryophyllene ozonolysis products and mixtures with commercially-available organic compounds

Young-Chul Song[1], Ariana G. Bé[2], Scot T. Martin[3], Franz M. Geiger[2], Allan K. Bertram[4], Regan J. Thomson[2], and Mijung Song[1*]

[1]Department of Earth and Environmental Sciences, Jeonbuk National University, Jeollabuk-do, Republic of Korea

[2]Department of Chemistry, Northwestern University, Evanston, Illinois 60208, United States

[3]School of Engineering and Applied Sciences & Department of Earth and Planetary Sciences, Harvard University, Cambridge, Massachusetts 02138, United States

[4]Department of Chemistry, University of British Columbia, Vancouver, BC, V6T 1Z1, Canada

*Correspondence: Mijung Song (mijung.song@jbnu.ac.kr)*

## Abstract

Liquid–liquid phase separation (LLPS) in organic aerosol particles can impact several properties of atmospheric particulate matter, such as cloud condensation nuclei (CCN) properties, optical properties, and gas-to-particle partitioning. Yet, our understanding of LLPS in organic aerosols is far from
complete. Here, we report on LLPS of one-component and two-component organic particles consisting of α-pinene- and β-caryophyllene-derived ozonolysis products and commercially-available organic compounds of relevance to atmospheric organic particles. In the experiments involving single-component organic particles, LLPS was observed in 8 out of 11 particle types studied. LLPS almost always occurred when the oxygen-to-carbon elemental ratio (O:C) was ≤ 0.44, but did not occur when
O:C was > 0.44. The phase separation occurred by spinodal decomposition, and when LLPS occurred, two liquid phases co-existed up to ~100% relative humidity (RH). In the experiments involving two-





component organic particles, LLPS was observed in 23 out of 25 particles types studied. LLPS almost always occurred when the average was O:C ≤ 0.67, but never occurred when the average O:C was > 0.67. The phase separation occurred by spinodal decomposition or growth of a second phase at the

surface of the particles. When LLPS occurred, two liquid phases co-existed up to ~100%. These results provide further evidence that LLPS is likely a frequent occurrence in organic aerosol particles in the troposphere, even in the absence of inorganic salts.

## 1. Introduction

Secondary organic aerosols (SOA) are ubiquitous in the atmosphere, comprising up to approximately 80% of the mass of submicrometer particles (Kanakidou et al., 2005; Jimenez et al., 2009; Heald et al., 2010). SOA particles are produced when OH, $NO_3$, and $O_3$ oxidize volatile organic compounds (VOC) in the atmosphere. Depending on the VOC type, oxidant type, and reaction time, the oxygen-to-carbon elemental ratio (O:C) of SOA can vary from 0.2 to 1.0 (Zhang et al., 2007; Hallquist et al., 2009;

Jimenez et al., 2009; Heald et al., 2010; Ng et al., 2010). SOA particles are important because they play critical roles in air quality, cloud formation, and the Earth's radiative properties (Seaton et al., 1995; Xiaohong and Jian, 2010; Pöschl and Shiraiwa, 2015; Sanchez et al., 2017; Shiraiwa et al., 2017).

SOA can undergo phase transitions as relative humidity (RH) changes in the atmosphere (Hänel, 1976; Martin, 2000; Krieger et al., 2012; You et al., 2014; Freedman, 2017). One possible phase

transition is liquid–liquid phase separation (LLPS) (Pankow, 2003; Marcolli and Krieger, 2006; Ciobanu et al., 2009; Bertram et al., 2011; Krieger et al., 2012; Song et al., 2012a; Zuend and Seinfeld, 2012; Veghte et al., 2014; You et al., 2014; Obrien et al., 2015; Freedman, 2017). The occurrence of LLPS has implications for the optical properties (Brunamonti et al., 2015; Fard et al., 2018), gas-particle partitioning (Zuend et al., 2010; Zuend and Seinfeld, 2012; Shiraiwa et al., 2013), hygroscopic

properties (Hodas et al., 2016), and cloud condensation nuclei (CCN) properties (Ovadnevaite et al., 2017; Liu et al. 2018) of atmospheric particles.

Many researchers have focused on LLPS in particles containing organic material mixed with inorganic salts. They found that LLPS can occur when the O:C of the organic material is smaller than 0.8 (Bertram et al., 2011; Krieger et al., 2012; Song et al., 2012a, 2012b; Schill and Tolbert, 2013; You


et al., 2013, 2014). More recently, studies on LLPS in organic aerosol particles free of inorganic salts have shown that LLPS occurs in SOA generated in environmental chambers when the average O:C of the organic material is smaller than roughly 0.5 across the RH range of ~95% to ~100% (Renbaum-Wolff et al., 2016; Rastak et al., 2017; Song et al., 2017; Ham et al., 2019) with implications for the CCN properties of the SOA (Petters et al., 2006; Hodas et al., 2016; Renbaum-Wolff et al., 2016;

Ovadnevaite et al., 2017; Rastak et al., 2017; Liu et al., 2018; Ham et al., 2019). Consistent with these laboratory studies, a recently introduced binary activity thermodynamic (BAT) model, with reduced complexity for atmospheric modelling, predicts that LLPS can occur when the O:C of the organic material is < 0.5 (Gorkowski et al., 2019), when considering different types of functional groups. In addition, Song et al. (2018) showed that LLPS occurs in organic particles containing one commercially-

available organic compound when the O:C is smaller than 0.44 while LLPS occurs in organic particles containing two commerically available organic species when the average the O:C is smaller than ≤ 0.58.

In the following, we investigated LLPS in particles containing one and two organic species generated from ozonolysis products of α-pinene and β-caryophyllene, which are atmospherically relevant, and commercially-available organic compounds. These results provide additional insight into the O:C range

required for LLPS in organic particles free of inorganic salts.

## 2. Experimental

### 2.1 Materials

Table 1 presents the physical properties of the organic compounds investigated. In this study, 11 organic

species were used, including seven products from the ozonolysis of α-pinene and β-caryophyllene and four commercially-available organic compounds. These species covered an O:C range of 0.13 - 1.00 (Table 1). All species were liquid at room temperature.

Seven of the products from the ozonolysis of α-pinene and β-caryophyllene were synthesized. The detailed synthesis methods for these species are described in Bé et al. (2017). Using [1]H NMR, [13]C NMR,

and IR spectroscopy, the ozonolysis products were characterized to confirm their identity and purity. All products contained a carboxylic acid, ketone, and/or aldehyde, which are abundant organic functional groups in the atmosphere (Hallquist et al., 2009; Nozière et al., 2015). The O:C range of the



ozonolysis products was between 0.13 and 0.44 (Table 1). To achieve O:C ratios up to 1.00, we used commercially-available organic compounds (Sigma-Aldrich, purities ≥ 97%) (Table 1).


## 2.2 Preparation of particles consisting of one and two organic species

Particles consisting of either one or two organic compounds were prepared at room temperature without the addition of a solvent. Particles consisting of the commercially-available organic compounds were nebulized directly on siliconized hydrophobic glass slides (Hampton Research, Canada). Particles

consisting of ozonolysis products were slightly viscous. To form particles on a substrate, these ozonolysis products were picked up with the tip of a pipette, and the pipette was then flicked towards a hydrophobic glass slide.

Particles consisting of two organic compounds were prepared using mixtures (1:1 mass ratio) of pure organic species without addition of a solvent. To prepare the mixtures with 1:1 mass ratio, each organic

species was weighed in a vial and then combined. After mixing, the solutions were homogenous based on visual inspection. Particles were generated from these mixtures and deposited on hydrophobic slides either by nebulization (for the mixtures involving commercially-available organic compounds) or by the flicking method via the tip of a pipette as described above (for the ozonolysis products). This method of producing two-component organic particles did not work for α-pinene ozonolysis products and β-

caryophyllinic acid due to the stickiness of these material. Hence, these materials were not included in the systems used to generate two-component organic particles.

## 2.3 Optical microscopy for observation of liquid–liquid phase separation

The organic particles on hydrophobic glass slides were placed into a RH and temperature controlled

flow-cell coupled to an optical microscope (Olympus BX43, 40× objective, Japan) (Parsons et al., 2004; Pant et al., 2006; Bertram et al., 2011; Song et al., 2012a, 2018; Ham et al., 2019). During all experiments, the temperature inside the flow-cell was maintained at 291 ± 1 K. The RH was controlled by a continuous flow of a wet and dry $N_2$ mixture with a total flow rate of 500 sccm. The temperature and RH were monitored by a humidity and temperature sensor (Sensirion, SHT 71, Switzerland). RH

inside the flow-cell was calibrated by measuring the deliquescence RH of four different pure inorganic



salts (potassium carbonate, sodium chloride, ammonium sulfate, and potassium nitrate) (Winston and Bates, 1960). The RH uncertainty from the calibration was ±1.5%.

At the beginning of LLPS experiments, organic particles inside the flow-cell were equilibrated at ~100% RH for 15–20 min. If LLPS was observed, the RH was decreased from ~100% to ~5-10% lower than the RH at which the two liquid phases merged into one phase followed by an increase in RH to ~100%. If LLPS was not observed, the RH was decreased from ~100% to ~0% RH, followed by an increase to ~100% RH. During all experiments, the RH was adjusted at a rate of $0.1 – 0.2\%$ RH min$^{-1}$. The optical images during experiments were recorded every 5 s using a CMOS (complementary metal–oxide–semiconductor) detector (DigiRetina 16, Tucsen, China). Organic particles were selected in the diameter range of 30–100 µm, which was required for LLPS experiments.

## 3. Results and discussion

### 3.1 Liquid–liquid phase separation in particles containing one organic species

Eleven different types of particles containing one organic species were investigated for LLPS at 291 ± 1 K. Out of the eleven different types of one-component organic particles studied, eight underwent LLPS during humidity cycles (Table S1). LLPS occurred in all one-component organic particles containing α-pinene and β-caryophyllene ozonolysis products.

Shown in Figure 1 and Movies S1-S7 are optical images recorded while the RH was decreased for all the cases where LLPS was observed in one-component organic particles. For these cases, two liquid phases were always observed at ~100% RH. As the RH was decreased, the two liquid phases merged into one liquid phase at ~95% RH, except for particles of β-caryophyllinic acid (Fig. 1e and Movie S5). For β-caryophyllinic acid particles, the two liquid phases merged into one liquid phase at 83.7% RH (Fig. 1e and Movie S5). Particles of β-caryophyllonic acid and β-nocaryophyllonic acid had a partially engulfed morphology after LLPS (Fig. 1b, d and Movies S2, S4) (Kwamena et al., 2010; Reid et al., 2011; Song et al., 2013) while the others particles had a core-shell morphology after LLPS. We expect that the inner phase consisted mainly of water while the outer phase consisted mainly of organic molecules because the amount of the inner phase reduced in size as the RH was decreased (Renbaum-Wolff et al., 2016; Song et al., 2017, 2018).



Shown in Figure 2 and Movies S8-S14 are optical images of the same seven particles shown in Fig. 1
and Movies S1-S7, except the images were recorded while the RH was increased, rather than decreased.
At low RH-values, the particles contained one phase. As the RH increased, LLPS occur at ~95% RH for
all cases exception for β-caryophyllinic acid particles, which underwent LLPS at 84.9% RH (Fig. 2e
and Movie S12). At the onset of LLPS, many small inclusions formed in the particles. As the RH was
further increased, the small inclusions coagulated and coalesced, and the particles continued to grow
(Fig. 2 and Movies S8-S14). At ~100% RH, all particles contained two liquid phases.

The mechanism for LLPS in the single-component organic particles was likely spinodal
decomposition based on the formation of many small inclusions at the onset of LLPS.  Spinodal
decomposition is a phase transition that occurs within a liquid without an energy barrier (Shelby, 1995;
Papon et al., 1999; Ciobanu et al., 2009; Song et al., 2012a). Previous studies also observed LLPS by
spinodal decomposition in α-pinene-derived SOA, β-caryophyllene-derived SOA, and limonene-derived
SOA (Renbaum-Wolff et al., 2016; Song et al., 2017; Ham et al., 2019).

Illustrated in Fig. 3a is the lower RH boundary for LLPS (LLPS$_{lower}$) and upper RH boundary for
LLPS (LLPS$_{upper}$) determined for one-component organic particles (blue symbols). LLPS occurred in
the one-component organic particles when the O:C was $\leq$ 0.44. Our results are consistent with the
results from Song et al. (2018), who observed LLPS in one-component organic particles when the O:C
was $\leq$ 0.44 (Fig. 3a, grey symbols). Our results are also consistent with LLPS$_{lower}$ and LLPS$_{upper}$
determined for SOA produced from α-pinene and β-caryophyllene (Renbaum-Wolff et al., 2016; Song
et al., 2017; Ham et al., 2019). In all cases, LLPS$_{upper}$ was ~ 100% RH.

The values of LLPS$_{lower}$ and LLPS$_{upper}$ determine in the current experiments using a decreasing RH
was within the uncertainty of LLPS$_{lower}$ and LLPS$_{upper}$ values determine in the experiments using an
increasing RH (Tables S1 and S2). In addition, no dependence on particle size was observed for
LLPS$_{lower}$ and LLPS$_{upper}$ within the size range investigated (30–100 μm).

Figure 4 shows the occurrence of LLPS in single-component organic particles as a function of O:C
and molar mass from the current study and as well as that of Song et al. (2018). For comparison
purposes, also included in Fig. 4 is the miscibility boundary of organic compounds based on the BAT
model (Gorkowski et al., 2019). Three of our measurements appear to disagree with the BAT model.



The small discrepancies between the current results and the predictions from the BAT model is likely related to the uncertainty in the BAT model (approximately an uncertainty of ±0.03 O:C units) and difference in organic compounds used in the current study and used to generate the BAT model. In the current study, we investigated compounds with more than one type of functional group while the BAT model used a single type of functional group to generate miscibility boundaries.

## 3.2 Liquid-liquid phase separation in particles containing two organic species

To better mimic the complexity of real aerosol compositions, we also studied LLPS in particles containing two organic species. Table S2 lists the 25 different mixtures investigated using combinations of β-caryophyllene ozonolysis products and commercially-available organic compounds. In total, 23 out of the 25 two-component organic particle types investigated underwent LLPS (Fig. 3b and Table S2). Shown in Fig. 5 and Movies S15-S19 are examples of images of two-component organic particles that underwent LLPS during a decrease in RH. Shown in Fig. 6 and Movies S20-S24 are the same five particles, but images recorded as the RH was increased.

Out of the 23 particles types that underwent LLPS, 19 of the particle types formed a core-shell morphology with decreasing RH. Only four particle types (β-caryophyllonic acid/suberic acid, β-caryophyllonic acid/polyethylene glycol-400(PEG-400), β-caryophyllene aldehyde/β-caryophyllonic acid, and β-caryophyllonic acid/β-nocaryophyllonic acid) formed a partially engulfed morphology with decreasing RH. As discussed in Sect. 3.1, the inner phase is expected to be mainly water while the outer phase is expected to be mainly organic material (Renbaum-Wolff et al., 2016; Song et al., 2017, 2018). As RH was decreased, the two liquid phases merged into one phase. For example, particles of β-caryophyllene aldehyde/PEG-400 merged into one phase at 39.9% RH (Fig. 5e and Movie S19).

In the experiments with two-component organic particles and increasing RH, in most cases (19 out of the 23 particle types that underwent LLPS), phase separation began with the abrupt formation of many small inclusions (e.g. Fig. 6a, b, e and Movies S20, 21, 24). This behavior suggests spinodal decomposition as the mechanism for LLPS. In contrast, in experiments with particles containing ozonolysis products mixed with pyruvic acid, phase separation began with the growth of a second phase at the surface of the particle as the RH increased (Figs. 6c, d and Movies S22, 23). This type of





mechanism was previously observed in organic/inorganic aerosol particles (Ciobanu et al., 2009; Song et al., 2012a).

Illustrated in Fig. 3b (blue symbols) is the lower RH boundary for LLPS (LLPS$_{lower}$) and upper RH boundary for LLPS (LLPS$_{upper}$) determined in the experiments with two-component organic particles. LLPS was observed in all cases when the average O:C ≤ 0.67. When LLPS was observe, LLPS$_{upper}$ was ~ 100% RH. These results are similar to previous results from Song et al. (2018) (gray symbols in Fig. 3b), even though they studied different types of two-component organic particles. Figure 3b also presents Sigmoid-Boltzmann fits of all data points from Song et al. (2018) and the current study to parameterize LLPS$_{lower}$ (solid line) and LLPS$_{upper}$ (dashed line) as a function of O:C. The parametrizations of Sigmoid-Boltzmann fits are given in the Supplement Information (Sect. S2).

## 4. Atmospheric implications

O:C of organic materials has been used to interpret and parameterize hygroscopicity (Jimenez et al., 2009), oxidation (Heald et al., 2010; Kroll et al., 2011), and mixing thermodynamics of organic aerosol particles (Donahue et al., 2011; Hodas et al., 2016). Previous studies have shown LLPS in mixed organic and inorganic aerosol particles often occurs for O:C < 0.8 (Bertram et al., 2011; Krieger et al., 2012; Song et al., 2012a, 2012b; Schill and Tolbert, 2013; You et al., 2013, 2014). Even in the absence of inorganic salts, the occurrence of LLPS was dependent on the O:C of organic materials (Renbaum-Wolff et al., 2016; Song et al., 2017; Song et al., 2018; Ham et al., 2019). Our results show that as compositional complexity increased from one organic species to two organic species, LLPS occurred over a wider range of average O:C values of organic materials (increasing from 0.44 to 0.67) (Figs. 3a and b). Considering the chemical complexity and the O:C ratio of organic particles in the troposphere (0.20 < O:C < 1.00) (Zhang et al., 2007; Hallquist et al., 2009; Jimenez et al., 2009; Heald et al., 2010; Ng et al., 2010), our result provided additional evidence that LLPS is likely a common feature of organic aerosols free of inorganic salts. A caveat is that the mixing ratio of 1:1 for two organic species and the chemical complexity used in our experiments is rather simple compared to the chemical complexity found in the atmosphere (Zhang et al., 2007; Hallquist et al., 2009; Jimenez et al., 2009).



Further studies are needed to confirm LLPS in organic aerosols comprising of more complex mixtures with different mixing ratios.

The occurrence of LLPS in organic aerosol particles at high RH, as observed in the current studies, is important since LLPS at high RH can lower the barrier to CCN activation by decreasing the surface tension of the particles (Ovadnevaite et al., 2017; Rastak et al., 2017; Liu et al., 2018). A decrease in surface tension and lowering of the barrier to CCN, can lead to an increase in cloud droplets numbers in the atmosphere, with implications for modelling the indirect effect of aerosols on climate (Ovadnevaite et al., 2017; Rastak et al., 2017).


Data availability. Underlying material and related items for this paper are located in the Supplement.

Author contributions. MS, AKB, and RJT designed the study. YS and MS conducted LLPS experiments
and analyzed the data. AGB, FMG, and RJT produced ozonolysis products. YS and MS prepared the manuscript with contributions from AGB, STM, FMG, AKB, and RJT.

Competing interests. The authors declare that they have no conflict of interest.

Acknowledgements. For authors at Jeonbuk National University, this work was supported by the National Research Foundation of Korea grant funded by the Korea Government (2019R1A2C1086187) and by Research Base Construction Fund Support Program funded from Jeonbuk National University in 2020. M. Song gives thanks to G. Jo for the technical support. The US National Science Foundation (AGS-1640378) is acknowledged by authors from Harvard University. The authors at Northwestern
University acknowledge support from the US National Science Foundation (CHE-1607640 and Graduate Research Fellowship to AGB).


Table 1. Molecular formula, molecular structure, molecular weight, oxygen-to-carbon elemental ratios (O:C), and functional groups of organic compounds studied. All compounds are liquid at room temperature.

| | Compounds | Molecular formula | Molecular structure | Molecular weight (g/mol) | O: C | Functional group |
|---|---|---|---|---|---|---|
| Ozonolysis products | β-caryophyllene aldehyde | $C_{15}H_{24}O_2$ | | 237.19 | 0.13 | Aldehyde, Ketone |
| | β-caryophyllonic acid | $C_{15}H_{24}O_3$ | | 252.35 | 0.20 | Carboxylic acid, Ketone |
| | β-nocaryophyllone aldehyde | $C_{14}H_{22}O_3$ | | 238.32 | 0.21 | Aldehyde, Ketone |
| | β-nocaryophyllonic acid | $C_{14}H_{22}O_4$ | | 254.32 | 0.29 | Carboxylic acid, Ketone |
| | β-caryophyllinic acid | $C_{14}H_{22}O_4$ | | 254 | 0.29 | Carboxylic acid |
| | Pinonaldehyde | $C_{10}H_{16}O_2$ | | 169.12 | 0.20 | Aldehyde, Ketone |
| | Pinic acid | $C_9H_{14}O_4$ | | 209.08 | 0.44 | Carboxylic acid, Ketone |
| Commercially-available organic compounds | Suberic acid monomethyl ester | $C_9H_{16}O_4$ | | 188 | 0.44 | Carboxylic acid, Ester |
| | Polyethylene glycol-400 | $C_{2n}H_{4n+2}O_{n+1}$ | | 400 | 0.56 | Alcohol, Ether |
| | Diethyl L-tartrate | $C_8H_{14}O_6$ | | 206 | 0.75 | Alcohol, Ester |
| | Pyruvic acid | $C_3H_4O_3$ | | 88.06 | 1.00 | Carboxylic acid, Ketone |

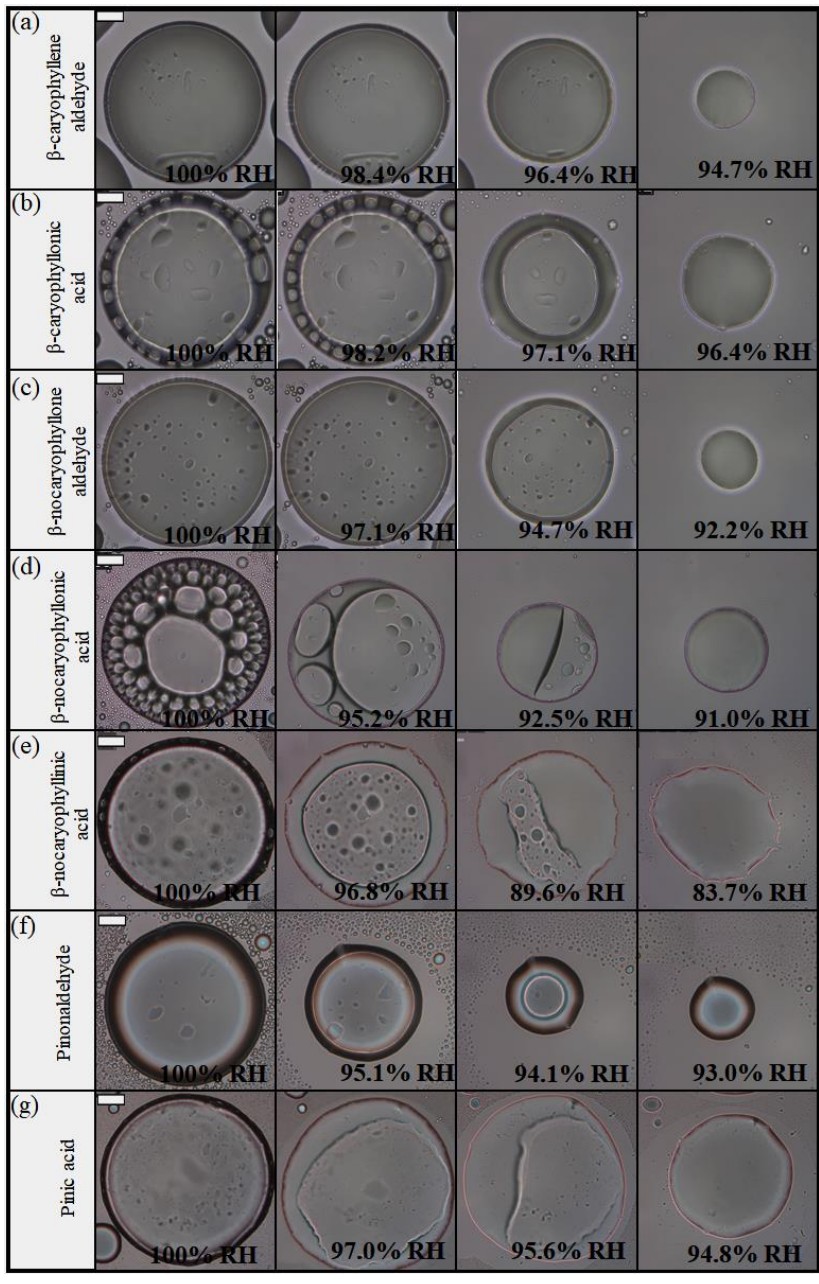

Figure 1. Optical images of particles for decreasing RH: (a) β-caryophyllene aldehyde, (b) β-caryophyllonic acid, (c) β-nocaryophyllone aldehyde, (d) β-nocaryophyllonic acid, (e) β-nocaryophyllinic acid, (f) pinonaldehyde, and (g) pinic acid. The last columns indicate the lower RH boundary for LLPS (LLPS$_{lower}$) with decreasing RH. The scale bar is 20 µm.



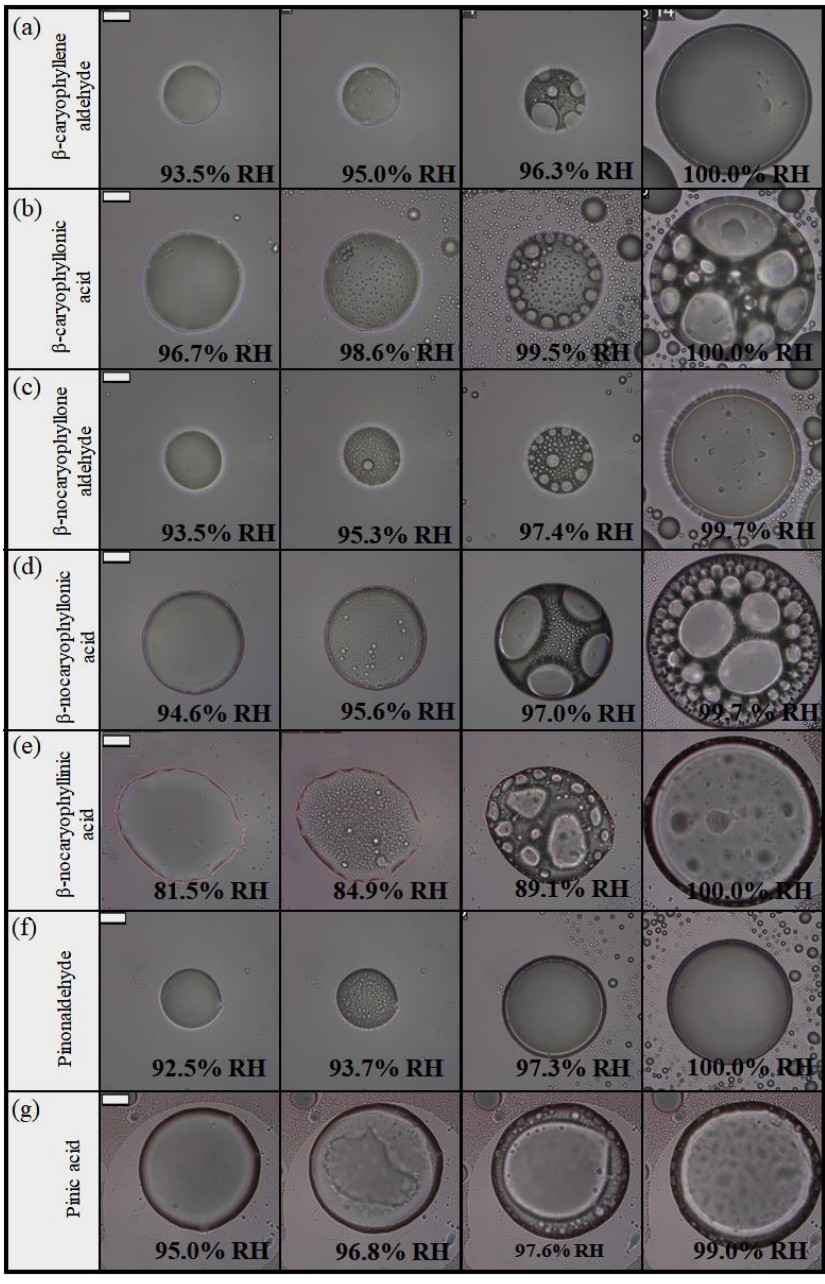

Figure 2. Optical images of particles for increasing RH: (a) β-caryophyllene aldehyde, (b) β-caryophyllonic acid, (c) β-nocaryophyllone aldehyde, (d) β-nocaryophyllonic acid, (e) β-nocaryophyllinic acid, (f) pinonaldehyde, and (g) pinic acid. The particles are the same ones in Fig. 1. The last columns indicate the upper RH boundary for LLPS (LLPS$_{upper}$) with increasing RH. The scale bar is 20 µm.



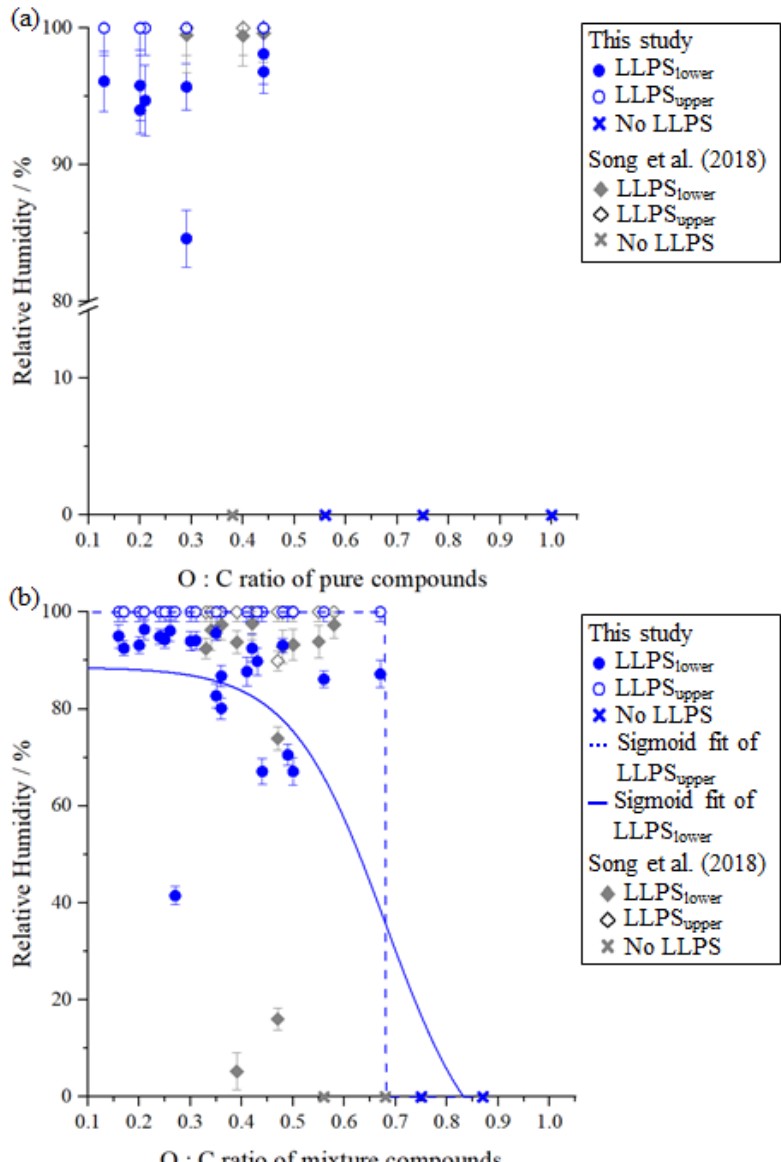

Figure 3. Relative humidity (RH) for LLPS as a function of the O:C of the organic particle consisting of: (a) one organic species, and (b) two organic species for increasing RH. Open blue circles are the LLPS upper boundary ($LLPS_{upper}$) with increasing RH, and closed blue circles are the LLPS lower boundary ($LLPS_{lower}$) with increasing RH. The grey diamonds are the result from Song et al. (2018). Error bars represent $2\sigma$ of multiple measurements and the uncertainty from the RH calibration. The solid and dashed lines are Sigmoid-Boltzmann fits to all the data of $LLPS_{lower}$ and $LLPS_{upper}$.



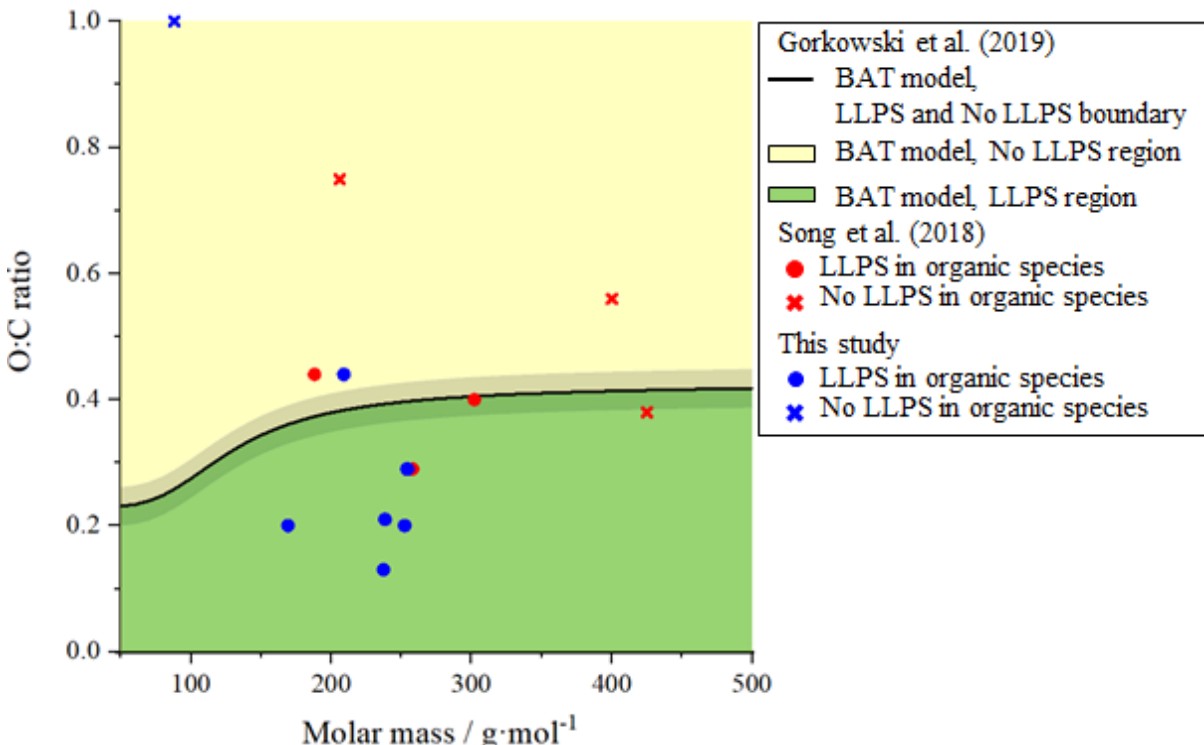

Figure 4. LLPS as a function of O:C and molar mass of particles of one organic species. Green and yellow regions indicate LLPS and no LLPS, respectively, in particles of one organic species using binary activity thermodynamics (BAT) model from Gorkowski et al. (2019). Black line indicates boundary of LLPS and no LLPS in the particles, and light gray shadow is the error region of the binary activity thermodynamic (BAT) model. Circles and crosses indicate measured LLPS and no LLPS, respectively, in one-component organic particles from Song et al. (2018) (red) and this study (blue).





Figure 5. Optical images of two-component particles for decreasing RH: (a) β-caryophyllene aldehyde/β-nocaryophyllonic acid, (b) Suberic acid monomethyl ester/pyruvic acid, (c) β-caryophyllonic acid/pyruvic acid, (d) β-nocaryophyllene aldehyde/pyruvic acid, and (e) β-caryophyllene aldehyde/polyethylene glycol-400. The last columns indicate the lower RH boundary for LLPS (LLPS$_{lower}$) with decreasing RH. The scale bar is 20 µm.





Figure 6. Optical images of two-component particles for increasing RH: (a) β-caryophyllene aldehyde/β-nocaryophyllonic acid, (b) Suberic acid monomethyl ester/pyruvic acid, (c) β-caryophyllonic acid/pyruvic acid, (d) β-nocaryophyllene aldehyde/pyruvic acid, and (e) β-caryophyllene aldehyde/polyethylene glycol-400. The particles are the same ones in Fig. 5. The last columns indicate the upper RH boundary for LLPS (LLPS$_{upper}$) with increasing RH. The scale bar is 20 µm.



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
