# Peer review of "Liquid–liquid phase separation and morphologies in organic particles consisting of $\alpha$ -pinene and $\beta$ -caryophyllene ozonolysis products and mixtures with commercially-available organic compounds"

_Atmospheric Chemistry and Physics, 2020_

## Referee Comment (RC1) · Nancy Lei (Referee) · 16 Apr 2020

The authors investigate liquid-liquid phase separation (LLPS) of $\alpha$-pinene and $\beta$-caryophyllene ozonolysis particles, as well as other atmospherically relevant organic particles. The different types of particles with different O:C ratios were studied under different relative humidity conditions. Results show that LLPS occurred to the single component organic particles with O:C smaller than 0.44, and for two-component organic particles with O:C smaller than 0.67. Overall, the results from this study present potential to improve current understanding of atmospherically relevant aerosol particles and with some revisions as noted below this should be publishable in Atmospheric Chemistry and Physics.

General Comments

The different types of particles with varying O:C show LLPS, the morphology (e.g. core-shell, engulf-coated, inclusions) of particles associated with different O:C ratios should be discussed more in details.

The figures show large size particles ∼80 to 100 $\mu$m, do smaller size particles (30 $\mu$m) present same result in this study?

Author expected the inner phase of the particle mainly consist of water and outer phase consisted mainly of organic. Would it be possible to use spectroscopy to confirm the chemical composition of the particle in different phases (e.g. core and shell)?

The inclusions present in different types of particles, which is very interesting and this occurred frequently in one-component particles. What are these inclusions and how does these inclusions form? Is this something correlated to O:C ratio?

Specific comments

Introduction

Line 57: How relevant the RH ranges (95%-100%) is compare to the atmospheric condition? Do you expect O:C ratio to be lower or higher to allow the LLPS occur at RH (50%-70%)?

Experimental

Line117: Does RH continuous decrease/increase? If so, how do you know if the particles reach to the equilibrium and the optical images is representative for that specific RH?

Line 120: Particle diameter of 30–100 $\mu$m seems like a big range, is this aerodynamic

diameter or the diameter after impaction? Why does author choose those sizes to study and how relevant comparing the particle sizes in the atmosphere?

Results and Discussion

Line 125: What caused other different types of particles not presenting LLPS?

Line 131: $\beta$-caryophyllinic acid and $\beta$-nocaryophyllonic acid has same O:C ratio but have very different behavior on LLPS, can author explain why?

Line 133: From Figure1g, the pinic acid at 95.6% RH also present engulf coated morphology, which is not included in discussion.

In Figure1, b, d, and e all exhibit inclusions inside the particle, what are these inclusions and how do they form?

Line 136: if the inner phase is considered as water and the outer phase are organic, do authors have data to support this statement? Most of the organic compounds are hydrophobic and wouldn't the outside organic layer prevent the water evaporation at lower RH?

Figure1f, the pinonaldehyde particles at 94.1% RH show multiple phase, looks like three layers, is that an artifact from microcopy image or that is real?

Figure 3, how many particles have been examined for each point? Do these particles has similar size? Previous study shows the size-dependent LLPS in atmospheric systems, which suggest smaller particles are likely present homogenous and large particles are likely to present LLPS. Could different sizes of particles in this study be a factor affect the results.

Line 185: It is very interesting to see the different mixture particles present LLPS at different RH, especially these two-component particles show LLPS at much lower RH. Can author explain what cause this? Is this can be triggered by high O:C ratio or large molecular weight of mixture particles?

---

## Referee Comment (RC2) · Anonymous Referee #2 · 7 May 2020

In this work, the authors provide new valuable experimental data related to the liquid-liquid phase separation (LLPS) of aqueous droplets containing single or two components found in the ozonolysis $\alpha$-pinene- and $\beta$-caryophyllene. The findings of works (e.g. relationship between LLPS and O/C) give us greater insights into the phase state of atmospheric aerosols under different environments, which largely govern many important atmospheric processes such as water uptake and CCN activities. I support the publication of this work and have some comments/suggestions for the authors' consideration.

Comments

In the introduction, the authors should provide more information why these classes of compounds are selected for this study. What are the atmospheric significances and abundances of these selected species? What the knowledge gap related to LLPS would like to be filled by investigating these compounds?

Page 3, Line 78, "Seven of the products from the ozonolysis of $\alpha$-pinene and $\beta$-caryophyllene were synthesized. The detailed synthesis methods for these species are described in Bé et al. (2017)." Please provide the purity of these synthesized chemicals used in this study.

Page 5, Line 113, "At the beginning of LLPS experiments, organic particles inside the flow-cell were equilibrated at ~100% RH for 15–20 min.". When the experiment ran at ~ 100%RH, does the condensation of water vapor on the surface of hydrophobic substrate and flow-cell affect the LLPS measurements?

Page 5, Line 119, "Organic particles were selected in the diameter range of 30–100 $\mu$m, which was required for LLPS experiments." Could these results be applicable to submicron sized aqueous droplets?

Page 5, Line 125, "Out of the eleven different types of one-component organic particles studied, eight underwent LLPS during humidity cycles (Table S1)." Could the authors comment how the chemical structure of the investigated compounds determine the occurrence of LLPS?

Page 5, Line 133, "Particles of $\beta$-caryophyllonic acid and $\beta$-nocaryophyllonic acid had a partially engulfed morphology after LLPS (Fig. 1b, d and Movies S2, S4) (Kwamena et al., 2010; Reid et al., 135 2011; Song et al., 2013) while the others particles had a core-shell morphology after LLPS." Could the authors comment how the chemical structure of the investigated compounds determine the morphology of the organic particles after LLPS?

[Figure]

Page 6, Line 164, "For comparison purposes, also included in Fig. 4 is the miscibility boundary of organic compounds based on the BAT model (Gorkowski et al., 2019)." Could the authors elaborate whether the BAT model can predict the RH at which LLPS occurs for the investigated compounds? Could the authors comment how the functional groups of the investigated compounds determine the occurrence of LLPS?

Page 7, Line 192, "In contrast, in experiments with particles containing ozonolysis products mixed with pyruvic acid, phase separation began with the growth of a second phase at the surface of the particle as the RH increased (Figs. 6c, d and Movies S22, 23)." Can the authors elaborate or explain this observation? What are the causes or mechanisms?

Atmospheric implications: can the authors further elaborate how different morphologies of the organic particles after LLPS affect atmospheric processes?

---

## Referee Comment (RC3) · Anonymous Referee #3 · 8 May 2020

General Comments:

In this manuscript, authors investigated the liquid-liquid phase separation (LLPS) as a function of average O:C ratio in organic particles free of inorganic species containing one component and binary mixture of α-pinene and β-caryophyllene-derived ozonolysis products and commercially available organic species. Compared to previous studies on this topic, this work used atmospherically relevant SOA products and showed that increased complexity of particulate organic species widen the range of O:C ratios over which LLPS will occur, improving our understanding of the LLPS behavior and

providing better constrain of the O:C range required for LLPS. I am supportive of the publication of this manuscript on Atmospheric Chemistry and Physics with the following comments/suggestions for the authors to consider in their revision.

Specific Comments: 1) Lines 163-171 and Figure 4: As indicated in the Gorkowski et al. (2019), the BAT model was intended for use to represent thermodynamics for with only bulk O:C information rather than a specific single organic system. It is not clear how the BAT model result was generated here. Is it simply a reproduction of the Figure 2 in the original paper (Gorkowski et al., 2019)? If it is, the comparison here doesn't seem to be fair. Or some modifications were made to tailor the model to the organic species studied in this work? If this is the case, could author include a section in the SI to describe the parameters and assumptions chosen when using that BAT model to generate the result shown in Figure 4? Either way, the discussion on Figure 4 doesn't seem to be sufficient. Could the author elaborate more on what implications one could draw from the discrepancies between the BAT model and observations? Especially if the model wasn't used in a system it was designed for the comparison here was potentially misleading. Given the complex composition and matrix effect within the ambient aerosols, it might be more appropriate to compare the observation vs. model comparison for the two component particles compared to one component particles.

2) Figure 3b showed that several points of LLPSlower RH were significantly lower than what the Sigmoid-Boltzmann fit would predict. It is obvious that O:C ratio is not a single determinant for LLPS. Authors should comment on possible explanations (relevant properties of the organic species, functional groups, spread in O:C values, etc.) for the variations of LLPSlower for two component organic particles.

Minor Comments: 1) On lines 132-133 $\beta$-caryophyllinic acid was discussed while the labeling on Figure 1e as well as in the caption was $\beta$-noncaryophyllininc acid.

2) It is hard to read the black texts of RH on top of the dark optical images. I would suggest either changing the color of the texts or not overlaying the labels and the

images.

3) Authors are recommended to double check the manuscript for grammatical errors. For example, on line 199, "When LLPS was observe" should be "When LLPS was observed".

---

## Author Comment (AC1) · 29 Jul 2020

We thank the referees for carefully reading our manuscript and for their helpful comments! Listed below are our responses to the comments from the referees of our manuscript. For clarity and visual distinction, the referee comments or questions are listed here in black and are preceded by bracketed, italicized numbers (e.g. *[1]*). Author's responses are offset in blue below each referee statement with matching numbers (e.g. *[A1]*).

**Response to Referee #1**

Summary: The authors investigate liquid-liquid phase separation (LLPS) of α-pinene and β-caryophyllene ozonolysis particles, as well as other atmospherically relevant organic particles. The different types of particles with different O:C ratios were studied under different relative humidity conditions. Results show that LLPS occurred to the single component organic particles with O:C smaller than 0.44, and for two-component organic particles with O:C smaller than 0.67. Overall, the results from this study present potential to improve current understanding of atmospherically relevant aerosol particles and with some revisions as noted below this should be publishable in Atmospheric Chemistry and Physics.

General Comments of Referee #1

*[1]* The different types of particles with varying O:C show LLPS, the morphology (e.g. core-shell, engulf-coated, inclusions) of particles associated with different O:C ratios should be discussed more in details.

*[A1]* Thank you for the comment. Different morphologies of core-shell, partially engulfed, and inclusions in the particles on the hydrophobic substrate were observed after LLPS occurred for RH increasing. These morphologies were also observed in the previous studies (Kwamena et al., 2010; Reid et al., 2011; Song et al., 2012a, b; 2013). The different morphologies of phase-separated particles deposited on the hydrophobic substrate can be explained by phase separation mechanisms, and variations in the volume ratio, chemical compositions, and spreading coefficients rather than the O:C ratios. Particles of β-nocaryophyllonic acid had inclusions after LLPS occurred, while other particles had a core-shell morphology after LLPS. Only one particle type (β-caryophyllonic acid/β-nocaryophyllonic acid) was observed both core-shell and partially engulfed morphology with increasing RH (Fig. S1). To address the referee's comment, we will discuss the morphologies in Sect 3.2 and present the morphologies of all particles studied in Tables S1 and S2. In addition, we will replace to the clearer optical images for β-caryophyllonic acid, β-nocaryophyllonic acid, β-nocaryophyllinic acid, pinonaldehyde and pinic acid (Figs. 1 and 2). Also, we will add an example of optical images for core-shell and partially engulfed morphologies of particles of β-caryophyllonic acid/β-nocaryophyllonic acid in Fig. S1. As the morphologies of the particles will be discussed more details in the revised manuscript, we will change the title of the manuscript to "Liquid–liquid phase separation and morphologies in organic particles consisting of α-pinene and β-caryophyllene ozonolysis products and mixtures with commercially-available organic compounds".

"Interestingly, particles showed different morphologies of core-shell, partially engulfed, and inclusions after LLPS occurred for RH increasing. These different morphologies have been also observed previously (Kwamena et al., 2010; Reid et al., 2011; Song et al., 2012a, 2013). The different morphologies could be resulted from variations in the phase separation mechanisms, volume ratios and different functional groups (dicarboxylic acid vs carboxylic acid and ketone) which can result in different interfacial energies and spreading coefficients (Kwamena et al., 2010; Reid et al., 2011; Song et al., 2013; Stewart et al., 2015, Gorkowski et al. 2020)."

References:

Kwamena, N. O. A., Buajarern, J., and Reid, J. P.: Equilibrium morphology of mixed organic/inorganic/aqueous aerosol droplets: Investigating the effect of relative humidity and surfactants, J. Phys. Chem. A., 114, 5787–5795, doi:10.1021/Jp1003648, 2010.

Reid, J. P., Dennis-Smither, B. J., Kwamena, N.-O. A., Miles, R. E. H., Hanford, K. L., and Homer, C. J.: The morphology of aerosol particles consisting of hydrophobic and hydrophilic phases: hydrocarbons, alcohols and fatty acids as the hydropho- bic component, Phys. Chem. Chem. Phys., 13, 15559–15572, doi:10.1039/C1CP21510H,doi:10.1039/C1CP21510H, 2011.

Song, M., Marcolli, C., Krieger, U. K., Zuend, A. and Peter, T.: Liquid-liquid phase separation and morphology of internally mixed dicarboxylic acids/ammonium sulfate/water particles, Atmos. Chem. Phys., 12, 2691–2712, doi:10.5194/acp-12-2691-2012, 2012a.

Song, M. J., Marcolli, C., Krieger, U. K., Lienhard, D. M., and Peter, T.: Morphologies of mixed organic/inorganic/aqueous aerosol droplets, Faraday Discuss., 165, 289–316, https://doi.10.1039/C3fd00049d, 2013

Stewart, D. J., Cai, C., Nayler, J., Preston, T. C., Reid, J. P., Krieger, U. K., Marcolli, C. and Zhang, Y. H.: Liquid-liquid phase separation in mixed organic/inorganic single aqueous aerosol droplets, J. Phys. Chem. A, 119(18), 4177–4190, doi:10.1021/acs.jpca.5b01658, 2015.

Gorkowski, K., Donahue, N. M. and Sullivan, R. C.: Aerosol Optical Tweezers Constrain the Morphology Evolution of Liquid-Liquid Phase-Separated Atmospheric Particles, Chem, 6(1), 204–220, doi:10.1016/j.chempr.2019.10.018, 2020.

*[2]* The figures show large size particles ～80 to 100 µm, do smaller size particles (30 µm) present same result in this study?

*[A2]* The result for LLPS in the organic particles was consistent within the size range of ~30 - ~100 µm in this study. To address the referee's comment, this information will be added to the revised manuscript (Sect. 3.1).

"These results for LLPS occurrence in the organic particles was consistent within the studied size ranges (~30-100 μm in diameter)."

*[3]* Author expected the inner phase of the particle mainly consist of water and outer phase consisted mainly of organic. Would it be possible to use spectroscopy to confirm the chemical composition of the particle in different phases (e.g. core and shell)?

*[A3]* Raman spectroscopy could be used to confirm the chemical composition of the phases, but this
technique was not available for the current study. We assume that the inner phase is mainly water and the outer phase is mainly organics because the amount of the inner phase reduced in size as the RH was decreased. This assumption, which is most likely valid, has also been used in several other studies (Renbaum-Wolff et al., 2016; Song et al. 2017, 2018; Ham et al. 2019). The surface tension of water and the surface tensions of organics are consistent with this assumption (Jasper, 1972). To address the
referee's comment, we will add the following to the revised manuscript (Sect. 3.1).

"We expect that the inner phase consisted mainly of water while the outer phase consisted mainly of organic molecules because the amount of the inner phase reduced in size as the RH was decreased (Renbaum-Wolff et al., 2016; Song et al., 2017, 2018). This assumption has also been reported in several
other studies (Renbaum-Wolff et al., 2016; Song et al. 2017, 2018; Ham et al. 2019). The surface tension of water and the surface tensions of organics are consistent with this assumption (Jasper, 1972).

References:
Ham, S., Babar, Z. Bin, Lee, J., Lim, H. and Song, M.: Liquid-liquid phase separation in secondary
organic aerosol particles produced from α-pinene ozonolysis and α-pinene photo-oxidation with/without ammonia, Atmos. Chem. Phys., 19(14), 9321–9331, doi:10.5194/acp-2019-19, 2019.
Jasper, J. J.: The surface tension of pure liquid compounds, J. Phys. And Chem. Ref. Data, vol 1, 841-1009, Doi: http://dx.doi.org/10.1063/1.3253106, 1972.
Renbaum-Wolff, L., Song, M., Marcolli, C., Zhang, Y., Liu, P. F., Grayson, J. W., Geiger, F. M., Martin,
S. T. and Bertram, A. K.: Observations and implications of liquid-liquid phase separation at high relative humidities in secondary organic material produced by α-pinene ozonolysis without inorganic salts, Atmos. Chem. Phys., 16(12), 7969–7979, doi:10.5194/acp-16-7969-2016, 2016.
Song, M., Liu, P., Martin, S. T. and Bertram, A. K.: Liquid-liquid phase separation in particles containing secondary organic material free of inorganic salts, Atmos. Chem. Phys., 17, 11261–11271,
doi:10.5194/acp-17-11261-2017, 2017.
Song, M., Ham, S., Andrews, R. J., You, Y. and Bertram, A. K.: Liquid-liquid phase separation in organic particles containing one and two organic species: importance of the average O:C, Atmos. Chem. Phys., doi:10.5194/acp-18-12075-2018, 2018.

*[4]* The inclusions present in different types of particles, which is very interesting and this occurred frequently in one-component particles. What are these inclusions and how does these inclusions form? Is this something correlated to O:C ratio?

*[A4]* The numerous small inclusions indicate onset of phase separation and are related to the mechanism of spinodal decomposition. Spinodal decomposition occurs with small inclusions within whole volume since there is no energy barrier to this type of phase transition (Ciobanu et al., 2009). The numerous small inclusions appeared and then coalesced on moistening resulting in inner and outer phases. The occurrence of LLPS was clearly correlated with O:C as shown in Fig. 3.

References:

Ciobanu, V. G., Marcolli, C., Krieger, U. K., Weers, U. and Peter, T.: Liquid-liquid phase separation in mixed organic/inorganic aerosol particles, J. Phys. Chem. A, 113(41), 10966–10978, doi:10.1021/jp905054d, 2009.

Specific comments of Referee #1

*[5]* Line117: Does RH continuous decrease/increase? If so, how do you know if the particles reach to the equilibrium and the optical images is representative for that specific RH?

*[A5]* At the beginning of LLPS experiments, the SOA particles were equilibrated at ~100% RH for 15-20 min. Then, RH in the flow-cell was continuously decreased and increased with ramp rates of 0.1 - 0.2 %

RH min$^{-1}$. We did not observe a dependence of LLPS and non-LLPS on the RH ramp rate. Previous studies of Song et al. (2017) and Ham et al. (2019), they used two RH rate: if LLPS was not observed, RH decreasing and increasing rates were 0.5–1.0% RH min$^{-1}$, and if LLPS was observed, they used 0.1–0.5% RH min$^{-1}$. In this case, they also did not observe a dependence of LLPS and non-LLPS on the different RH ramp rate. In the current study, we use a slower range of ramp rates. Moverover, particle size did not change at a specific RH.

References:

Song, M., Liu, P., Martin, S. T. and Bertram, A. K.: Liquid-liquid phase separation in particles containing secondary organic material free of inorganic salts, Atmos. Chem. Phys., 17, 11261–11271,
doi:10.5194/acp-17-11261-2017, 2017.
Ham, S., Babar, Z. Bin, Lee, J., Lim, H. and Song, M.: Liquid-liquid phase separation in secondary organic aerosol particles produced from α-pinene ozonolysis and α-pinene photo-oxidation with/without ammonia, Atmos. Chem. Phys., 19(14), 9321–9331, doi:10.5194/acp-2019-19, 2019.

*[6]* Line 120: Particle diameter of 30–100 µm seems like a big range, is this aerodynamic diameter or the diameter after impaction? Why does author choose those sizes to study and how relevant comparing the particle sizes in the atmosphere?

*[A6]* This is a good question! In this study, supermicrometer-sized particles (30-100 µm) were used since this is an optimum size to reach good quality images using the optical microscope equipped with a long working distance objective. We did not observe a size dependence for the LLPS within the size ranges studied. So far, there is no study investigated size effects for LLPS in organic particles free of inorganic salts. However, using organic/inorganic aerosol particles, Krieger et al. (2012) and You et al. (2014) reported that the separation relative humidity is not significantly depended on the micrometer-sized particles. Altaf et al. (2016), and Altaf and Freedman (2017) showed that LLPS was independent of size down to ~50 nm when a slow drying rate was used. These results can support that the phase separation behaviour with the supermicron particles provides organic particles that dominated the aerosols in the accumulation mode in large parts of the troposphere.

References:

Altaf, M. B., Zuend, A. and Freedman, M. A.: Role of nucleation mechanism on the size dependent morphology of organic aerosol, Chem. Commun., 52(59), 9220–9223, doi:10.1039/c6cc03826c, 2016.

Altaf, M. B. and Freedman, M. A.: Effect of Drying Rate on Aerosol Particle Morphology, J. Phys. Chem. Lett., 8(15), 3613–3618, doi:10.1021/acs.jpclett.7b01327, 2017.

Krieger, U. K., Marcolli, C. and Reid, J. P.: Exploring the complexity of aerosol particle properties and processes using single particle techniques, Chem. Soc. Rev., 41(19), 6631–6662, doi:10.1039/C2CS35082C, 2012.

You, Y., Smith, M. L., Song, M., Martin, S. T. and Bertram, A. K.: Liquid-liquid phase separation in atmospherically relevant particles consisting of organic species and inorganic salts, Int. Rev. Phys. Chem., 33(1), 43–77, doi:10.1080/0144235X.2014.890786, 2014.

*[7]* Line 125: What caused other different types of particles not presenting LLPS?

*[A7]* We showed that the oxygen-to-carbon elemental (O:C) ratio of the organic compounds is an important parameter to predict occurrence of LLPS and absence of LLPS. In single-component organic particles, LLPS almost always occurred when the O:C was $\leq 0.44$, but did not occur when the O:C was $>$

0.44. In two-component organic particles, LLPS almost always occurred when the average was O:C $\leq$ 0.67, but never occurred when the average O:C was $> 0.67$.

*[8]* Line 131: β-caryophyllinic acid and β-nocaryophyllonic acid has same O:C ratio but have very different behavior on LLPS, can author explain why?

*[A8]* Both particles of β-caryophyllinic acid and β-nocaryophyllonic acid occurred LLPS. Particles of β-caryophyllinic acid occurred LLPS by a mechanism of spinodal decomposition while particles of β- nocaryophyllonic acid occurred LLPS by a mechanism of nucleation and growth. Due to the different phase separation mechanisms, the two particles exhibited different morphologies. Regarding the particle morphologies, please see response to *[A1]*.

*[9]* Line 133: From Figure 1g, the pinic acid at 95.6% RH also present engulf coated morphology, which is not included in discussion. In Figure1, b, d, and e all exhibit inclusions inside the particle, what are these inclusions and how do they form?

*[A9]* Thank you for the comment. To clarify the morphologies, we will replace to better optical images of Fig. 1. Particles of pinic acid have core-shell morphology rather than engulf coated morphology on a hydrophobic substrate. We expect that the inclusions with a core-shell morphology consisted mainly of water as discussed above in *[A3]*. Such morphology with inclusions has been observed (Ciobanu et al., 2009; Song et al., 2012a, 2013)

References:

Ciobanu, V. G., Marcolli, C., Krieger, U. K., Weers, U. and Peter, T.: Liquid-liquid phase separation in mixed organic/inorganic aerosol particles, J. Phys. Chem. A, 113(41), 10966–10978, doi:10.1021/jp905054d, 2009.

Song, M., Marcolli, C., Krieger, U. K., Zuend, A. and Peter, T.: Liquid-liquid phase separation and morphology of internally mixed dicarboxylic acids/ammonium sulfate/water particles, Atmos. Chem. Phys., 12, 2691–2712, doi:10.5194/acp-12-2691-2012, 2012a.

Song, M. J., Marcolli, C., Krieger, U. K., Lienhard, D. M., and Peter, T.: Morphologies of mixed organic/inorganic/aqueous aerosol droplets, Faraday Discuss., 165, 289–316, https://doi.10.1039/C3fd00049d, 2013

*[10]* Line 136: if the inner phase is considered as water and the outer phase are organic, do authors have data to support this statement? Most of the organic compounds are hydrophobic and wouldn't the outside organic layer prevent the water evaporation at lower RH?

*[A10]* Please see response to *[A3]* above. We did not observe significant water evaporation of organic layer at certain RH.

*[11]* Figure 1f, the pinonaldehyde particles at 94.1% RH show multiple phase, looks like three layers, is that an artifact from microcopy image or that is real?

*[A11]* It is an optical artifact in Fig. 1f. In the revised manuscript, we will replace to the clearer images for a pinonaldehyde particle.

*[12]* Figure 3, how many particles have been examined for each point? Do these particles has similar size? Previous study shows the size-dependent LLPS in atmospheric systems, which suggest smaller particles are likely present homogenous and large particles are likely to present LLPS. Could different sizes of particles in this study be a factor affect the results.

*[A12]* In Fig. 3, each data point is included 4~5 particles within the size range of 30 to 100 µm. We will add this information in the revised manuscript (Sect. 2.3 and Fig. 3 caption). We did not observe a size-dependent within this size range. A more detailed response is in *[A2]*.

*[13]* Line 185: It is very interesting to see the different mixture particles present LLPS at different RH, especially these two-component particles show LLPS at much lower RH. Can author explain what cause this? Is this can be triggered by high O:C ratio or large molecular weight of mixture particles?

*[A13]* This is a good question! As shown in Fig. 3b, the value of the $LLPS_{lower}$ tends to decrease as the O:C ratio increases. However, organic functional groups could also be an important parameter for LLPS that still need to be studied. To address the referee's comments, we will add the following discussion (Sect. 4):

"In addition to the O:C ratio, the types of organic functional groups present in the molecules are also likely important for LLPS (Song et al., 2012b) because different functional groups lead to different strengths of intermolecular interactions with water. Further studies are needed to elucidate the effect of functional groups on the occurrence of LLPS in organic particles."

References:
Song, M., Marcolli, C., Krieger, U. K., Zuend, A. and Peter, T.: Liquid-liquid phase separation in aerosol particles: Dependence on O:C, organic functionalities, and compositional complexity, Geophys. Res. Lett., 39, L19801, doi:10.1029/2012GL052807, 2012b.

---

## Author Comment (AC2) · 29 Jul 2020

We thank the referees for carefully reading our manuscript and for their helpful comments! Listed below are our responses to the comments from the referees of our manuscript. For clarity and visual distinction, the referee comments or questions are listed here in black and are preceded by bracketed, italicized numbers (e.g. [1]). Author's responses are offset in blue below each referee statement with matching numbers (e.g. [A1]).

5

**Response to Referee #2**

Summary: In this work, the authors provide new valuable experimental data related to the liquid-liquid phase separation (LLPS) of aqueous droplets containing single or two components found in the ozonolysis 10  $\alpha$ -pinene- and  $\beta$ -caryophyllene. The findings of works (e.g. relationship between LLPS and O/C) give us greater insights into the phase state of atmospheric aerosols under different environments, which largely govern many important atmospheric processes such as water uptake and CCN activities. I support the publication of this work and have some comments/suggestions for the authors' consideration.

15

**Comments**

[1] In the introduction, the authors should provide more information why these classes of compounds are selected for this study. What are the atmospheric significances and abundances of these selected species? What the knowledge gap related to LLPS would like to be filled by investigating these compounds?

- [A1] Thank you for the comment.  $\alpha$ -pinene and  $\beta$ -caryophyllene are the most abundant types of 20 monoterpene ( $C_{10}H_{16}$ ) and sesquiterpenes ( $C_{15}H_{24}$ ) in the atmosphere, respectively (Guenther, 1995; Pathak et al., 2007; Henrot et al., 2017). However, the studies of LLPS and morphologies for  $\alpha$ -pinene and  $\beta$ -caryophyllene oxidation products are still rare. Compared to previous studies on LLPS in organic particles, we investigated atmospherically relevant SOA products and showed that increased complexity
- 25 of particulate organic species widen the range of O:C ratios over which LLPS will occur, improving our understanding of the LLPS behavior and providing better constrain of the O:C range required for LLPS. To address the referee's comment, we will add the following to the introduction of the revised manuscript (Sect. 1).
- " $\alpha$ -pinene and  $\beta$ -caryophyllene are the most abundant types of monoterpene (C10H16) and sesquiterpenes 30  $(C_{15}H_{24})$  in the atmosphere, respectively (Guenther, 1995; Sakulyanontvittaya et al. 2008; Henrot et al., 2017). However, the studies of LLPS and morphologies for  $\alpha$ -pinene and  $\beta$ -caryophyllene oxidation products are still rare. Our results can provide additional insight into the O:C range required for LLPS in organic particles free of inorganic salts. Moreover, our results can provide that the chemical complexity
- of organic particles effects on LLPS. These observations should improve our understanding of LLPS 35 behavior and provide more accurate constrained the value of the O:C ratio for LLPS. The results from

these studies should also improve the understanding and modelling of CCN activity of SOA free of inorganic salts."

[revised manuscript text omitted]

---

## Author Comment (AC3) · 29 Jul 2020

We thank the referees for carefully reading our manuscript and for their helpful comments! Listed below are our responses to the comments from the referees of our manuscript. For clarity and visual distinction, the referee comments or questions are listed here in black and are preceded by bracketed, italicized numbers (e.g. *[1]*). Author's responses are offset in blue below each referee statement with matching numbers (e.g. *[A1]*).

**Response to Referee #3**

Summary: In this manuscript, authors investigated the liquid-liquid phase separation (LLPS) as a function of average O:C ratio in organic particles free of inorganic species containing one component and binary mixture of α-pinene and β-caryophyllene-derived ozonolysis products and commercially available organic species. Compared to previous studies on this topic, this work used atmospherically relevant SOA products and showed that increased complexity of particulate organic species widen the range of O:C ratios over which LLPS will occur, improving our understanding of the LLPS behavior and providing better constrain of the O:C range required for LLPS. I am supportive of the publication of this manuscript on Atmospheric Chemistry and Physics with the following comments/suggestions for the authors to consider in their revision.

*[1]* Specific Comments: 1) Lines 163-171 and Figure 4: As indicated in the Gorkowski et al. (2019), the BAT model was intended for use to represent thermodynamics for with only bulk O:C information rather than a specific single organic system. It is not clear how the BAT model result was generated here. Is it simply a reproduction of the Figure 2 in the original paper (Gorkowski et al., 2019)? If it is, the comparison here doesn't seem to be fair. Or some modifications were made to tailor the model to the organic species studied in this work? If this is the case, could author include a section in the SI to describe the parameters and assumptions chosen when using that BAT model to generate the result shown in Figure 4? Either way, the discussion on Figure 4 doesn't seem to be sufficient. Could the author elaborate more on what implications one could draw from the discrepancies between the BAT model and observations? Especially if the model wasn't used in a system it was designed for the comparison here was potentially misleading. Given the complex composition and matrix effect within the ambient aerosols, it might be more appropriate to compare the observation vs. model comparison for the two component particles compared to one component particles.

*[A1]* For the miscibility gap in Fig 4 in the manuscript, we used the miscibility line from Fig. 2a and SI in Gorkowski et al. (2019). As stated by the referee and the authors of the BAT model, the point of the BAT model is to represent the bulk O:C and molar mass dependences for mixtures, and the BAT model may not represent well the thermodynamics of a single organic system. Nevertheless, we thought that it was interesting that the BAT model was reasonably consistent with our measurements. The referee

suggests that this comparison may be unfair, and their criticism is reasonable.  As a result, to address the referee's comments, we will remove the comparison between our results and the BAT model (Fig. 4) in the revised manuscript.

40

The referee also suggested that a comparison between the two component particles and the BAT model may be more appropriate.  This is true, except if the O:C values of the two organic components vary greatly.  In this case, the low O:C component in the mixture can remain immiscible and the high O:C component can remain miscible for a wide range of RH values.  Unless we misunderstand, the BAT model was not developed to describe this type of situation. To avoid additional concerns that we are using the BAT model outside the conditions intended, we would prefer not to compare the results for the two component particles with the BAT model.

45

50 *[3]* Figure 3b showed that several points of $LLPS_{lower}$ RH were significantly lower than what the Sigmoid-Boltzmann fit would predict. It is obvious that O:C ratio is not a single determinant for LLPS. Authors should comment on possible explanations (relevant properties of the organic species, functional groups, spread in O:C values, etc.) for the variations of $LLPS_{lower}$ for two component organic particles.
*[A3]* As the referee commented, three data of the $LLPS_{lower}$ of one mixture of polyethylene glycol-400/
55 β-caryophyllene aldehyde from this study and two mixtures of polyethylene glycol-400/diethyl sebacate and polyethylene glycol-400/glyceryl tributyrate from Song et al. (2018) were lower than what the Sigmoid-Boltzmann fit. We will add the discussion to address the comment (Sect. 4).

"In addition to the O:C ratio, the types of organic functional groups present in the molecules are also
60 likely important for LLPS (Song et al., 2012b) because different functional groups lead to different strengths of intermolecular interactions with water. Further studies are needed to elucidate the effect of functional groups on the occurrence of LLPS in organic particles."

References:

65 Song, M., Marcolli, C., Krieger, U. K., Zuend, A. and Peter, T.: Liquid-liquid phase separation in aerosol particles: Dependence on O:C, organic functionalities, and compositional complexity, Geophys. Res. Lett., 39, L19801, doi:10.1029/2012GL052807, 2012b.
Song, M., Ham, S., Andrews, R. J., You, Y. and Bertram, A. K.: Liquid-liquid phase separation in organic particles containing one and two organic species: importance of the average O:C, Atmos. Chem. Phys.,
70 doi:10.5194/acp-18-12075-2018, 2018.

Minor Comments:

*[4]* On lines 132-133 β-caryophyllinic acid was discussed while the labeling on Figure 1e as well as in the caption was β-noncaryophyllininc acid.

*[A4]* Thank you for the correction. We will correct the Figure 1 in the manuscript.

*[5]* It is hard to read the black texts of RH on top of the dark optical images. I would suggest either changing the color of the texts or not overlaying the labels and the images.

*[A5]* As suggested, we will revise the color and font size of the Figures 1, 2, 5, and 6.

*[6]* Authors are recommended to double check the manuscript for grammatical errors. For example, on line 199, "When LLPS was observe" should be "When LLPS was observed".

*[A6]* Thank you for the correction. We will correct them!